# The Financial Market of Indices of Socioeconomic Well-Being

Thilini V. Mahanama [1], Abootaleb Shirvani [2,*], Svetlozar Rachev [3] and Frank J. Fabozzi [4]

1  Department of Industrial Management, University of Kelaniya, Kelaniya 11600, Sri Lanka; thilinim@kln.ac.lk
2  Department of Mathematical Science, Kean University, Union, NJ 07083, USA
3  Department of Mathematics & Statistics, Texas Tech University, Lubbock, TX 79409, USA; zari.rachev@ttu.edu
4  Carey Business School, Johns Hopkins University, Baltimore, MD 21218, USA; ffabozz1@jhu.edu
*  Correspondence: ashirvan@kean.edu

**Abstract:** This study discusses how financial economic theory and its quantitative tools can be applied to create socioeconomic indices and develop a financial market for the so-called "socioeconomic well-being indices". In this study, we quantify socioeconomic well-being by assigning a dollar value to the well-being factors of selected countries; this is analogous to how the Dow 30 encapsulates the financial health of the US market. While environmental, social, and governance (ESG) financial markets address socioeconomic issues, our focus is broader, encompassing national citizens' well-being. The dollar-denominated socioeconomic indices for each country can be viewed as financial assets that can serve as risky assets for constructing a global index, which, in turn, serves as a "market of well-being socioeconomic index". This novel global index of well-being, paralleling the Dow Jones Industrial Average (DJIA), provides a comprehensive representation of the world's socioeconomic status. Through advanced financial econometrics and dynamic asset pricing methodologies, we evaluate the potential for significant downturns in both the socioeconomic well-being indices of individual countries and the aggregate global index. This innovative approach allows us to engineer financial instruments akin to portfolio insurance, such as index puts, designed to hedge against these downturn risks. Our findings propose a financial market model for well-being indices, encouraging the financial industry to adopt and trade these indices as mechanisms to manage and hedge against downturn risks in well-being.

**Keywords:** socioeconomic well-being indices; dynamic asset pricing theory; world development socioeconomic indicators; global dollar socioeconomic well-being index; gross domestic product; financial econometric modeling

## 1. Introduction

A country's policymakers require a measure of the magnitude of their economic output to evaluate the impact of economic policies on growth. Since 1934, the gross national product (GNP), proposed by Simon Kuznets, has served as the primary metric in the US and other developed countries, encompassing all final transactions of goods and services. In 1991, the gross domestic product (GDP), a similar measure, took the spotlight. However, the GNP/GDP faced criticism for focusing solely on aggregate economic wealth, neglecting individual citizens' economic welfare and well-being. Leaders of major countries, like then-President of France Nicolas Sarkozy in 2008, recognized these limitations, leading to the identification of diverse sources of well-being beyond economic indicators that could be used to assess not just economic performance but also social progress.[1]

The IMF's report discusses income, consumption, and wealth distribution indicators for categorizing well-being and economic welfare (Quiros-Romero and Reinsdorf 2020). Ahmad and Qureshi (2021) highlight the need for an index measuring citizen well-being, allowing policymakers to benchmark their performance. This approach is exemplified by the United Nations Development Programme's Human Development Index (HDI), which evaluates various dimensions of human development beyond economic growth (Sagar and

Najam 1998). The HDI represents achievements in health, knowledge, and living standards. The OECD has also pioneered economic indicators to comprehensively assess economic and social progress, such as the "Better Life Index", which reflects a country's current socioeconomic status based on factors like housing, income, education, and governance (OECD 2023).

Trindade et al. (2020) developed a socioeconomic well-being index using the historical mood of the US population, issuing marketable financial contracts like options and futures through financial econometric modeling and dynamic asset pricing theory. These indices offer early warnings for policymakers and private agents regarding potential mood-related market downturns. However, a country's well-being is not solely dependent on the public mood or political factors; it should also encompass the role of the business community.

Our study develops a conceptual framework for encapsulating socioeconomic well-being; it is comparable to the way the Dow 30 indices encapsulate the industrial and financial vitality of corporations in the US. In this paper, we take a global view of a nation's well-being, applying dynamic asset pricing theory and integrating financial economic theory to assess and manage the risks associated with socioeconomic indices, expanding on prior work on environment, society, and governance (ESG) and socially responsible investing (Matos 2020).

Our strategy involves establishing a financial market for socioeconomic well-being indices, treating them as risky financial assets, and assigning dollar values by normalizing these assets between 0 and 100; this is akin to ESG scores. We construct socioeconomic well-being indices for nine countries: the US, Australia, Brazil, China, Germany, India, Japan, South Africa, and the UK. Incorporating eight key world development indicators—the Gini index, unemployment, life expectancy at birth, gross national income, inflation rate, population, and foreign direct investment from the World Bank (IBRD 2022)—we create dollar-denominated indices, utilizing a country's GDP to determine their financial value. These dollar-denominated indices are derived to measure and assign a tangible monetary value to the well-being factor, establishing a precedent in the financial analysis of social metrics.

We establish a market for dollar-denominated socioeconomic well-being indices, a *global index* akin to the Dow Jones Industrial Average (DJIA), treating them as risky financial assets. Our global index provides an equally weighted market index, with each country's dollar-denominated socioeconomic well-being index considered a risky asset within this composite index. Leveraging these indices from developed countries, we conduct optimization analyses to derive efficient frontiers. We analyze the stability and risks associated with the socioeconomic well-being indices of individual nations and the global composite index by employing financial econometrics and dynamic asset pricing theory. We delve into the likelihood and potential impact of sharp downturns in these indices, which reflect broader socioeconomic challenges and trends.

By integrating financial theory with well-being metrics, we create a platform for financial instruments analogous to those used for hedging against market volatility, such as portfolio insurance. These instruments, such as index puts, are designed to mitigate the risk of downturns in the indices of well-being for countries and the global index alike. This concept opens up a new market for securities that protect against declines in social welfare, much like how traditional financial instruments protect against economic downturns. For financial markets, this represents an expansion of tradable assets to include indices that represent human and social capital. For policymakers and social scientists, it offers a new lens through which to view the economic implications of social policies and changes. For the global community, it emphasizes the importance of socioeconomic factors as integral elements of a nation's wealth and prosperity.

Our analysis applies portfolio theory, complying with Basel II Accord requirements, and employs various risk–return measures. This paper discusses the development of a market where these socioeconomic well-being indices are traded as derivatives, providing a hedging mechanism against the socioeconomic risks faced by countries. The financial instruments that we propose are similar in nature to index puts, which allow investors to

insure their portfolios against declines. We anticipate that institutional investors will trade these assets similarly to other derivatives. For instance, a US equity portfolio manager forecasting a decline in the US dollar-denominated socioeconomic well-being index may opt for portfolio insurance instruments or short positions in the index. In the context of our socioeconomic well-being indices, these instruments would serve to protect against downturns in a nation's socioeconomic conditions. The availability of such instruments could also facilitate a more proactive approach to social investment and policymaking, encouraging a forward-looking stance on well-being.

Existing well-being indices lack dynamic features; they lack a time series econometric model that can be used to forecast and trade them in financial portfolios. No research has explored financial instruments for assessing a nation's downturns. Thus, we establish a market for socioeconomic indices, enabling the financial industry to engage in societal well-being assessment and the management of potential downturns. While ESG plays a vital role in addressing socioeconomic concerns in financial markets, our aim is broader, encompassing a nation's overall well-being.

During the preparation of this work, the authors utilized ChatGPT 3.5 and ChatGPT 4 for the sole purpose of enhancing the linguistic quality of our manuscript. After employing these AI tools, the authors meticulously reviewed and edited the con-tent as needed, thereby taking full responsibility for the content of the publication. The use of these AI resources was strictly limited to improving the clarity and coherence of the text, given the diverse writing styles of the four authors. At no point was AI used for any analytical input or data analysis within the paper.

The rest of this paper unfolds as follows: in Section 2, we construct dollar-denominated well-being indices for nine countries using a mix of indicators that reflect economic, social, and developmental factors. Then, we normalize these indices, allowing them to be compared and aggregated into a global index that offers a holistic view of the global well-being. In Sections 3 and 4, we assess the stability and potential volatility of these indices using dynamic asset pricing, and we present the optimal portfolio composition and efficient frontier in Section 5. Section 6 introduces an option pricing model for the global socioeconomic well-being index, and this is followed by our concluding remarks in Section 7.

## 2. World Development Socioeconomic Well-Being Indicators

The World Bank (2022) compiles internationally comparable statistics on global development and poverty alleviation. This section outlines the world development indicators used to construct our USD-denominated socioeconomic index. Our index primarily incorporates indicators related to income, health, labor, education, the economy, and global connections. These include the Gini index for income distribution, total population, life expectancy at birth, and unemployment, which measure population health, labor, and education dynamics. The gross national income (GNI), consumer price index (CPI), and GDP measure income, savings, prices, the terms of trade, and the economic structure. Foreign direct investment reflects financial flows related to global connections. Further descriptions of these world development indicators (IBRD 2022) can be found in the online appendix. Whenever it is necessary, we compute missing data using multiple imputations with principal component analysis (Josse et al. 2011). We modify the world development indicators to construct the global dollar socioeconomic well-being index in Section 2.1.

### 2.1. Constructing a Global Dollar Socioeconomic Well-Being Index

In this subsection, we construct a global socioeconomic well-being index using the eight world development indicators (WDIs) defined by the World Bank (Gini index, total unemployment, life expectancy at birth, GNI per capita, consumer price index, population, foreign direct investment (in current USD), GDP/capita) for the nine countries in our study. Based on the data availability in IBRD (2022), we selected these nine countries from all the continents and extracted reported data from between 1990 and 2020.

We denote by $F(k,l)$ the $k$th world development indicator ($k = 1, \ldots, K = 8$) for the $l$th country ($l = 1, \ldots, L = 9$) for a given year such that all indicators are strictly positive; that is, $F(k,l) > 0$ for all $k,l$ so that they positively contribute to the socioeconomic well-being index. Therefore, we transform the Gini index and unemployment indicators into the neg. Gini index (100-Gini index) and employment (100-unemployment), respectively. The US development indicators that positively contribute to the US socioeconomic well-being index are shown in Figure 1a.

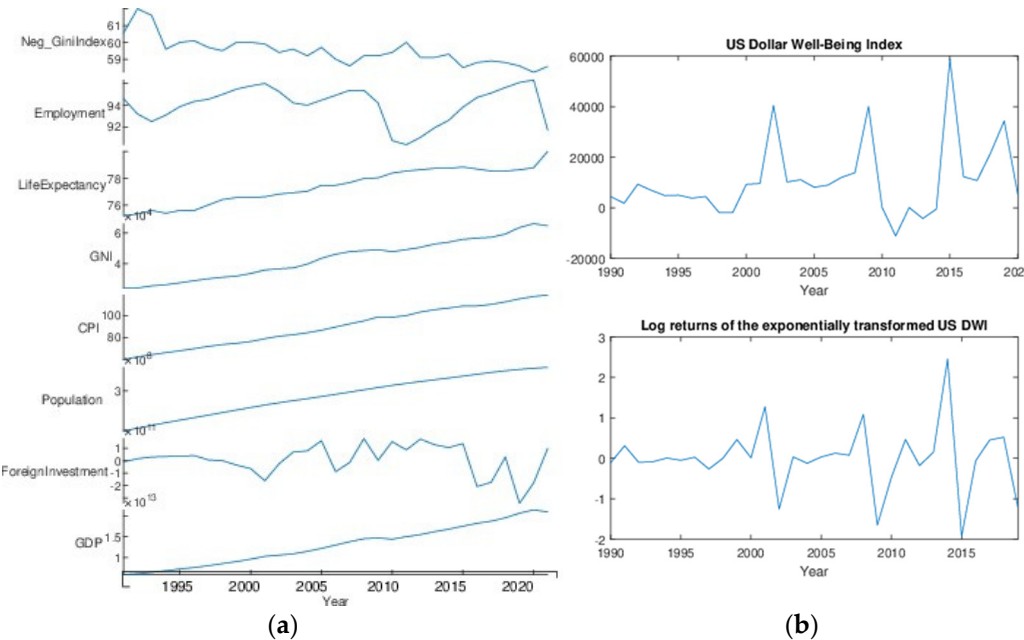

(a)          (b)

**Figure 1.** (**a**) US development indicators from World Bank reports (IBRD 2022) for 1990−2020 and (**b**) the US dollar socioeconomic well-being index (DWI) constructed using Equation (3) and the log returns of the exponentially transformed DWI with constraints from Equation (5).

In order to compare the significance of factors with respect to countries, we normalize each country's indicator, $F(k,l)$, using the corresponding indicator of all the countries ($F(k,l)$, $l = 1, \ldots, L = 9$) as follows:

$$FN(k,l) = \frac{F(k,l)}{\sum_{l=1}^{L} F(k,l)}, \; k = 1, \ldots, K, \; l = 1, \ldots, L. \tag{1}$$

We then define the socioeconomic well-being index for country $l$, which we denote by $WI(l)$, as the average of its normalized indicators, excluding the GDP:

$$WI(l) = \frac{1}{K-1} \sum_{k=1}^{K-1} FN(k,l), \; l = 1, \ldots, L, \tag{2}$$

so that $WI(l) \in (0,1)$, $l = 1, \ldots, L$. We monetize the US dollar value of $WI(l)$ at year $t$, $WI_t(l)$, by weighting it with its corresponding GDP/capita, $GDP_t(l)$, to define a US dollar-denominated index of socioeconomic well-being, the dollar socioeconomic well-being index (DWI), for country $l$ at year $t$, as follows:

$$DWI_t(l) = GDP_t(l) \cdot WI_t(l), \; t = t_0 = 1990, \ldots, t = 2020. \tag{3}$$

This reflects the "well-being" of a resident in country $l$ in US dollars. For instance, Figure 1b depicts the US per capita well-being. Figure 2a highlights variations in the well-being of different countries based on the dollar value per capita (DWI).

To compare the well-being of an individual with that of the overall population, we construct a global DWI by taking the average of the DWIs:

$$DWI_t = \frac{1}{L} \sum_{l=1}^{L} DWI_t(l), \ t = t_0, \ldots, t_{30}. \tag{4}$$

The global DWI is shown in Figure 2b. The yearly deviations in terms of (thousands of) USD represent the change in the DWI per person from one year to another.

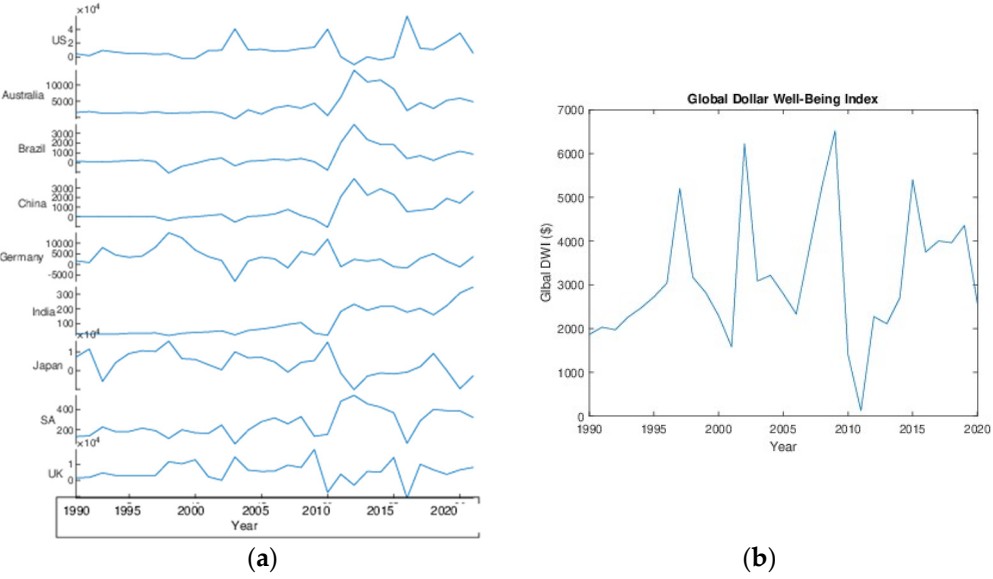

**(a)**        **(b)**

**Figure 2.** (**a**) Dollar socioeconomic well-being indices constructed using Equation (3) and (**b**) the global DWI proposed in Equation (4).

### 2.2. Econometric Financial Modeling of Socioeconomic Well-Being Indices

To model population well-being within a financial framework, we need to assign a US dollar value to the well-being of each resident in a given country, treating it as an asset price. This approach involves applying dynamic asset pricing theory to capture the financial market dynamics of these socioeconomic well-being indices over time. The exponential transformation of these time series is selected to mirror asset price dynamics, facilitating the application of dynamic asset pricing theory (Duffie 2010; Schoutens 2003).

We determine the exponential transformation for each DWI, considering the years $t = t_0, \ldots, 2020$, as follows:

$$f(x) = a\exp(bx), \ a > 0, \ b > 0,$$
$$f\left(\min_{l,t} DWI_t(l)\right) = 0, \tag{5}$$
$$f\left(\max_{l,t} DWI_t(l)\right) = 1.$$

Under this optimization, we set the exponential transformation of the lowest DWI to 0 and that of the highest DWI to 1. We assume that the "asset price" (the happiness of the representative inhabitant) should have a minimal value of 0 and a maximal value of 1, which represents 100%. This scale is the one used in ESG rankings (Scatigna et al. 2021). The exponential transformation for the US DWI produces $a = 0.0037$ and $b = 0.0001$. With all DWIs being positive, we define the log returns for the exponentially transformed DWI and thus introduce the well-being asset pricing model for each country with respect to the global index.

We take the log returns of the exponentially transformed DWIs in the way that this is done in dynamic asset pricing theory (Duffie 2010):

$$R_t(l) = log \frac{f(DWI_t(l))}{f(DWI_{t-1}(l))}; l = 1, \ldots, 10, \ t = t_0, \ldots, t_{30}, \quad (6)$$

where $f(DWI_t(l))$, $l = 1, \ldots, 9$, is the exponentially transformed DWI for the $l$th country in year $t$, and we set $l = 10$ for the exponentially transformed global DWI. In each country, we model the log returns of the exponentially transformed DWI with an autoregressive AR(1) model:

$$R_t(l) = \phi_0 + z_t + \theta_1 z_{t-1}, \ t = t_0, \ldots, t_{30}, \quad (7)$$

where $z_t = \sigma_t \epsilon_t$, and $\epsilon_t$ are assumed to be independent and identically distributed (iid) innovations, while $\phi_0$ and $\theta_1$ are new parameters to be estimated. We model the volatility ($\sigma_t$) using the best fit from among the time-varying volatility models ARCH(1), GARCH(1,1), and EGARCH(1,1). The GARCH(1,1) model is defined as

$$\sigma_t = \frac{z_t}{\epsilon_t},$$
$$\sigma_t^2 = \alpha_0 + \alpha_1 z_{t-1}^2 + \beta_1 \sigma_{t-1}^2, t = t_1, \ldots, t_{30}, \quad (8)$$

where $\phi_0$, $\theta_1$, $\alpha_0$, $\alpha_1$, and $\beta_1$ are parameters to be estimated (Bollerslev 1986). The sample innovations, $\epsilon_t$, are iid random variables with zero mean and unit variance (Tsay 2005).

We model the log returns $R_t(l)$ in Equation (7) using the following univariate models with standard normal iid innovations (Hamilton 2020; Tsay 2005):

- Model 1: AR(1)-ARCH(1);
- Model 2: AR(1)-GARCH(1,1);
- Model 3: AR(1)-EGARCH(1,1).

We compare the performances of the three models based on the Akaike information criterion (AIC) and the Bayesian information criterion (BIC), as shown in Table 1. For each country, we select the model that results in the lowest AIC and BIC among the estimated models (Models 1, 2, and 3). For example, Model 1 outperforms the other models in modeling the log returns of the US DWI. Combining the simulated sample innovations of each country, we get a 10-dimensional sample of 30 observations. We employ $S = 10,000$ scenarios for innovations based on a fitted 10-dimensional normal-inverse Gaussian (NIG) distribution, which is in the domain of attraction of a 10-dimensional multivariate Gaussian distribution (Øigård et al. 2004). By passing from normal to NIG innovations, we preserve the asymptotic unbiasedness of the parameters in the marginal time series models (Equation (8)). To assess the confidence bound for these parameters, we would need bootstrapping methods that are beyond the scope of this paper. Next, using the estimated parameters in Models 1, 2, and 3 for each of the 10 marginal (one-dimensional) time series, we generate $S = 10,000$ Monte Carlo scenarios of the innovations[2] for the log returns in the year 2021 (($R_{t31}(1; s), \ldots, R_{t31}(10; s)$), $s = 1, \ldots, S$). As a result, we obtain $S$ scenarios for the log returns of the 10 indices for the year 2021. As the innovations are models with an NIG distribution, which captures the tail dependencies of the indices, our overall forecast of the socioeconomic market for 2021 exhibits all the "stylized facts" of a financial market (see Taylor 2011; Cont 2001).

We use the simulated scenarios and the estimated parameters of the estimated univariate time series model (AR(1)-ARCH(1) or AR(1)-GARCH(1,1) or AR(1)-EGARCH(1,1)) to forecast $S$ dynamic log returns for each DWI for 2021 ($R_{2021}$). For example, the estimated model for the US DWI log returns is an AR(1)-EGARCH(1,1) (with multivariate NIG sample innovations) model with the following parameters:

$$R_t(l) = 0.09 + z_t - 0.13 z_{t-1},$$
$$\sigma_t = \frac{z_t}{\epsilon_t},$$
$$\sigma_t^2 = -2.97 + 0.95 z_{t-1}^2 - 0.93 \sigma_{t-1}^2, \ t = t_0, \ldots, t_{30}, \quad (9)$$

where the $\epsilon_t$ are iid random normal innovations.

**Table 1.** Estimated dynamic model comparison for the log returns of DWIs in Equation (6) based on the AIC and BIC (Model 1: AR(1)-ARCH(1), Model 2: AR(1)-GARCH(1,1), Model 3: AR(1)-EGARCH(1,1)).

| Country | Model 1 | AIC Model 2 | Model 3 | Model 1 | BIC Model 2 | Model 3 |
|---------|---------|-------------|---------|---------|-------------|---------|
| US | 2.4627 | 2.5209 | 2.5960 | 2.6495 | 2.7544 | 2.8498 |
| Australia | 1.5981 | 0.8630 | 0.5255 | 1.7849 | 1.0965 | 0.8058 |
| Brazil | 1.6695 | 1.7362 | 1.4492 | 1.8563 | 1.9697 | 1.7294 |
| China | 1.8168 | 1.4721 | 1.4353 | 1.0037 | 1.7056 | 1.7156 |
| Germany | 2.2366 | 2.3033 | 2.1957 | 2.4235 | 2.5368 | 2.4760 |
| India | 0.9228 | 0.9895 | 0.5342 | 1.1097 | 1.2230 | 0.8144 |
| Japan | 2.7870 | 2.8536 | 2.4123 | 2.9738 | 3.0872 | 2.6926 |
| SA | 2.1488 | 2.2155 | 2.1473 | 2.3357 | 2.4490 | 2.4276 |
| UK | 2.5711 | 2.6378 | 2.4302 | 2.7580 | 2.8713 | 2.7105 |
| Global DWI | 2.4346 | 2.5013 | 2.4220 | 2.6214 | 2.7348 | 2.7022 |

Then, we simulate $S$ dynamic log returns for 2021, which we use as the dynamic asset prices for the US. In summary, we simulate dynamic log returns (for the year 2021, t31 = 2021) for each country by estimating econometric models for their historical log returns (between 2000 and 2020). The joint dependence of the log returns of all indices is determined by a multivariate NIG distribution on the sample innovations. Using the historical and dynamic econometric models of the indices as time series, we provide an asset valuation risk analysis. Note that historical time series analyses cannot be used for dynamic asset pricing, particularly for option pricing. Our econometric models are designed to be consistent with dynamic asset pricing theory, allowing the valuation of the indices as financial assets and the pricing of financial contracts, particularly insurance instruments, on the socioeconomic well-being indices.

## 3. Measuring the Tail Risk of Socioeconomic Well-Being Indices

In this section, we assess the DWI's performance using economic factors linked to adverse changes in each country's index, specifically evaluating its response to adverse events using the eight world development indicators. The Ljung–Box[3] test indicates serial correlation and dependence in historical log returns. Our analysis emphasizes capturing linear and nonlinear return dependencies through dynamic log returns, which yield stationary time series. Initially, we fit multivariate NIG models to each country's dynamic log returns (iid standardized innovations). Subsequently, we generate 10,000 scenarios from the multivariate NIG for scenario analysis and systemic risk measure computation.

To assess the tail risk, we calculate three systemic risk measures derived from the value at risk (VaR) (Philippe 2001). We calculate the VaR at the quantile levels of $(1 - \alpha)100\%$, with $\alpha = 0.05$ and $\alpha = 0.01$. For a given $\alpha$-quantile level ($0 < \alpha < 1$), $VaR_\alpha$ is defined as follows:

$$VaR_\alpha(X) = -\inf\{x \epsilon R | F_X(x) > \alpha\},\ 0 < \alpha < 1, \tag{10}$$

where $F_X(x)$ is the cumulative distribution function of the log return $X$.

Tobias and Brunnermeier (2016) proposed a $\Delta$-conditional value at risk (CoVaR) measure, which in our setting represents the log return of a DWI; it shows how the VaR of the financial system changes when a financial institution experiences distress relative to its median state. We use the conditional value at risk (CVaR) (Rockafellar and Uryasev 2000) to find the tail risk in the log returns of the DWI, denoted by $X$, at the $\alpha$ levels $(1 - \alpha)100\%$ = 95% and $(1 - \alpha)100\%$ = 99%, as follows:

$$CVaR_\alpha(X) = \frac{1}{\alpha} \int_0^\alpha VaR_\gamma(X) d\gamma. \tag{11}$$

If $X$ has a probability density, then the CVaR coincides with the expected tail loss (ETL) or expected shortfall (ES) as follows (Rachev et al. 2008):

$$CVaR_\alpha(X) = -E(X|X \leq -VaR_\alpha(X)). \tag{12}$$

By switching from the VaR to the CVaR, the latter measure being a coherent risk measure,[4] we may take into account episodes of more severe distress, back-test (Acerbi and Szekely 2014) the CVaR, and enhance its monotonicity concerning the dependence parameter. Girardi and Ergun (2013) improved the method for identifying whether a financial institution is considered to be in financial distress, from being exactly at its $VaR_\alpha(X)$ ($X = -VaR_\alpha(X)$) to being less than or equal to its $VaR_\alpha(X)$ ($X \leq -VaR_\alpha(X)$). Here, we refer to $Y$ as the global DWI log returns and $X$ as the log returns of the indices for each country in the alternative CoVaR defined in terms of the copulas in Mainik and Schaanning (2014). The CoVaR at the level of $\alpha$, $CoVaR_\alpha$ (or $\xi_\alpha$), is defined as

$$\xi_\alpha = CoVaR^\alpha = -F^{-1}_{Y|X \leq F^{-1}_X(\alpha)}(\alpha) = -VaR_\alpha(Y|X \leq -VaR_\alpha(X)), \tag{13}$$

where $F_Y$ and $F_X$ denote the cumulative distributions of $Y$ and $X$, respectively, and $F_{Y|X}$ is the cumulative conditional distribution of $Y$ given $X$. An extension of CoVaR, the conditional expected shortfall (CoES) for DWI log returns (Mainik and Schaanning 2014) at a level $\alpha$, is given by

$$CoES_\alpha = -E(Y|Y \leq -\xi_\alpha, X \leq -VaR_\alpha(X)). \tag{14}$$

The conditional expected tail loss (CoETL) is the average of the DWI losses when the DWI and the country indices' extreme indicators are in distress (Biglova et al. 2014). The CoETL quantifies the portfolio downside risk in the presence of systemic risk. We define the CoETL at a level $\alpha$ as follows:

$$CoETL_\alpha = -E(Y|Y \leq -VaR_\alpha(Y), X \leq -VaR_\alpha(X)). \tag{15}$$

We assess the portfolio market risk by computing systemic risk measures (VaR, CoVaR, CoES, and CoETL) for the joint densities of the global DWI and individual country DWIs, as presented in Table 2. These values gauge the impact of a significant increase in each country's DWI on the global DWI. We choose confidence levels of 95% and 99% for statistical comparisons and disastrous loss calculations. The first column displays empirical correlation coefficients (Pearson's $R$), revealing no linear relationship between Germany and the DWI. Strong positive linear relationships exist between the DWI and the US and Japan. Conversely, China, Brazil, and South Africa (SA) exhibit robust negative relationships with the DWI, accompanied by low Pearson's R values that indicate weak relationships.

Columns 2 and 3 in Table 2 present each country's DWI VaR measures, which represent the maximum losses at the 95% and 99% confidence levels. These losses have 5% and 1% chances of surpassing the VaR threshold, respectively. Notably, the US experiences the maximum loss at the 99% confidence level for both dynamic and historical log returns. At the 95% confidence level, the highest loss is observed in the US (2.88) for dynamic data, while the UK records the highest loss (3.13) in historical data.

We measure the average loss beyond the 5% and 1% confidence levels using CVaR, which represents the expected loss in each country's worst 1% and 5% scenarios. Notably, the US registers the maximum average losses at both the 95% and 99% confidence levels. The CoES and CoETL assess how a significant decrease in each country's DWI affects the global DWI. For instance, at the 5% stress level, the CoES signifies the expected return on the DWI in the top 5% of each country's index. Remarkably, CoES5% and CoETL5% across all countries are quite similar, suggesting comparable impacts of a drastic DWI increase in each country on the global DWI. However, in the most severe scenario (1%), where the

stress on the US DWI has the greatest influence on the global DWI, Australia exhibits the lowest impact on the global DWI for dynamic log returns.

In summary, Table 2 presents the market risk based on left-tail systemic risk measures for the global DWI concerning each country's DWI at varying stress levels (5% and 1%). These insights can aid investors in evaluating the global DWI market risk and optimizing their portfolios in the global financial market of socioeconomic well-being indices.

**Table 2.** Comparison of Pearson's *R* and left-tail systemic risk measures (VaR, CoVaR, CoES, and CoETL) of the joint densities of the global DWI and DWI of each country at different stress levels for dynamic and historical log returns.

| Country | Pearson's R | VaR %95 | VaR %99 | CVaR %95 | CVaR %99 | CoES %95 | CoES %99 | CoETL %95 | CoETL %99 |
|---|---|---|---|---|---|---|---|---|---|
| **Dynamic Log Return Left-Tail Risk Measures** | | | | | | | | | |
| US | 0.55 | 2.88 | 2.55 | 3.56 | 3.66 | 0.43 | 0.61 | 1.54 | 2.65 |
| Australia | −0.25 | 0.52 | 0.39 | 0.52 | 0.66 | 0.47 | 0.65 | 0.31 | 0.53 |
| Brazil | −0.63 | 0.96 | 0.52 | 0.70 | 1.24 | 0.47 | 0.65 | 0.59 | 0.96 |
| China | −0.76 | 0.76 | 0.40 | 0.62 | 1.08 | 0.47 | 0.65 | 0.36 | 0.76 |
| Germany | 0.00 | 1.31 | 0.39 | 0.58 | 1.70 | 0.46 | 0.63 | 0.55 | 1.22 |
| India | −0.41 | 2.11 | 1.10 | 1.39 | 2.89 | 0.47 | 0.65 | 1.06 | 2.12 |
| Japan | 0.40 | 0.60 | 0.26 | 0.47 | 0.75 | 0.45 | 0.63 | 0.28 | 0.58 |
| SA | −0.58 | 1.53 | 0.84 | 1.11 | 2.00 | 0.47 | 0.65 | 0.89 | 1.54 |
| UK | 0.24 | 0.67 | 0.37 | 0.47 | 0.80 | 0.45 | 0.64 | 0.43 | 0.66 |
| **Historical log return left-tail risk measures** | | | | | | | | | |
| US | 0.79 | 2.74 | 4.87 | 4.30 | 5.66 | 1.85 | 3.83 | 1.16 | 4.70 |
| Australia | −0.65 | 1.81 | 3.14 | 2.91 | 3.46 | 2.44 | 3.97 | 1.93 | 3.15 |
| Brazil | −0.79 | 1.81 | 3.47 | 3.19 | 3.86 | 2.44 | 3.97 | 1.97 | 3.49 |
| China | −0.82 | 2.09 | 3.73 | 3.36 | 4.23 | 2.44 | 3.97 | 2.18 | 3.75 |
| Germany | 0.04 | 2.35 | 2.80 | 2.67 | 2.97 | 2.44 | 3.97 | 2.35 | 2.80 |
| India | −0.72 | 1.42 | 2.64 | 2.34 | 3.05 | 2.44 | 3.97 | 1.47 | 2.65 |
| Japan | 0.57 | 2.77 | 2.85 | 2.83 | 2.87 | 2.44 | 3.97 | 2.54 | 2.85 |
| SA | −0.77 | 2.28 | 3.69 | 3.42 | 4.06 | 2.44 | 3.97 | 2.39 | 3.70 |
| UK | −0.06 | 3.13 | 4.57 | 4.24 | 5.02 | 2.09 | 3.97 | 2.70 | 4.37 |

## 4. Regression and Jensen's Alphas of the Socioeconomic Well-Being Indices with Respect to the Global Socioeconomic Well-Being Index

This section describes the estimation of the relationships between a country's DWI and the global DWI. We use a regression analysis to capture the pairwise linear dependences between each DWI and the global DWI as follows:

$$Y_l = a_l + b_l Y + e_l, \tag{16}$$

where $Y_l$ and $Y$ are the log returns of the exponentially transformed DWI of country $l$ and the global index, respectively. The terms $a_l$, $b_l$, and $e_l$ denote the intercept, gradient, and random error corresponding to the regression line, respectively.

The ordinary least squares (OLS) method is typically used to estimate the parameters $a_l$ and $b_l$ when the errors follow a normal distribution. However, this assumption breaks down when there are outliers and highly influential observations. Robust regression (RR) addresses this issue by iteratively reweighting least squares to assign optimal weights to each data point (Hu et al. 2021; Knez and Ready 1997). OLS is suitable for short-term predictions, as it excludes outliers, while RR is preferred for longer-term forecasts.

We apply RR and OLS to the log returns of each DWI and the global DWI. For instance, we compare the regression analysis performance for the US DWI using data observed from 1990 to 2020 (historical regression) with the data generated for 2021 using time series modeling (dynamic regression), as discussed in Section 2.2. This involves regressing over $S = 10,000$ Monte Carlo scenarios for both individual and global indices.[5] Table 3 presents estimated coefficients (intercept and gradient) for the regression models, along with goodness-of-fit measures (*p*-values, standard errors, and root mean square error (RMSE)). In dynamic RR and OLS models, the estimated coefficients are quite similar to

those in the historical models. However, dynamic regression models outperform historical ones, exhibiting lower *p*-values and standard errors. Moreover, dynamic RR surpasses dynamic OLS, with lower RMSEs and standard errors for the estimated coefficients.

Figure 3 shows the estimated regression lines for historical and dynamic data. The historical data, which are from 1990–2020, represent the average indicator values used in constructing the index, while the dynamic data comprise 10,000 generated scenarios for 2021. Historical regression is ineffective due to heavy-tailed asymmetry and volatility clustering in the data. Conversely, dynamic data conform to the normal distribution, as discussed in Section 2.2. The dynamic model captures autoregression, asymmetric volatilities, and heavy-tailed asymmetric copula dependence. Given increasing volatility over time, the dynamic model is preferable due to volatility clustering.

**Table 3.** Regression analysis for historical and dynamic log returns in the US.

| Data Type | | Regression Type | | | |
|---|---|---|---|---|---|
| | | RR | | | |
| | | Coefficient | *p*-Value | Standard Error | RMSE |
| Historical | Intercept (a) | −0.026 | 0.705 | 0.070 | 0.381 |
| | Gradient (b) | 0.673 | 0.000 | 0.093 | |
| Dynamic | Intercept (a) | 0.275 | 0.000 | 0.007 | 0.489 |
| | Gradient (b) | 1.313 | 0.000 | 0.017 | |
| | | OLS | | | |
| Historical | Intercept (a) | −0.008 | 0.937 | 0.093 | 0.512 |
| | Gradient (b) | 0.862 | 0.000 | 0.125 | |
| Dynamic | Intercept (a) | 0.378 | 0.000 | 0.008 | 0.603 |
| | Gradient (b) | 1.360 | 0.000 | 0.022 | |

Both historical and dynamic regressions for the US DWI yield upward forecasts (refer to Figure 3). Addressing the grim state of global well-being is essential. A steeper gradient in the dynamic regression signifies a more promising future well-being for a country. Table 4 presents estimated gradients from dynamic RR for all countries in the study. Among the considered countries, the US boasts the highest well-being, while South Africa ranks lowest. Detailed regression analyses for all countries are available from the authors.

**Table 4.** Robust regression for historical and dynamic log returns in the US. Regression lines for both historical and dynamic data result in upward forecasts.

| Country | Estimated Gradient | Standard Error |
|---|---|---|
| US | 1.36 | 0.0174 |
| Japan | 0.11 | 0.0025 |
| UK | 0.07 | 0.0031 |
| Germany | 0.01 | 0.0125 |
| Australia | −0.22 | 0.0092 |
| Brazil | −0.77 | 0.0095 |
| China | −1.04 | 0.0087 |
| India | −1.11 | 0.0025 |
| SA | −1.14 | 0.0157 |

In the log return time series of the indices, negative drops are prominent, and these downturns are effectively captured by the NIG distribution of innovations (Schlösser 2011, pp. 129–63). Hence, we employ AR(1)-ARCH(1,1) or AR(1)-EGARCH(1,1) with a multivariate NIG distribution for option pricing, providing flexible probability distributions that account for heavy tails and dependencies, both centrally and in the tail. Dynamic models enable us to assess the value of insurance instruments accurately. Historical (static) methods fall short in providing suitable pricing models for socioeconomic well-being indices; they lack the ability to establish no-arbitrage values for insurance instruments. Thus, we must rely on no-arbitrage asset pricing theory and implement dynamic models

for pricing the well-being of individual countries and the global index. Therefore, we employ dynamic predictive models to forecast future trends in the global well-being that are country-specific and on a global scale.

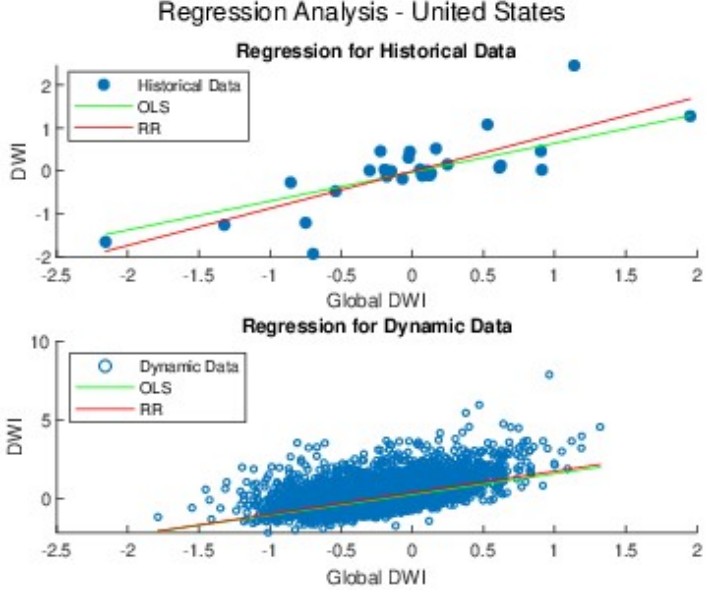

**Figure 3.** Robust regression for historical and dynamic log returns in the US. Regression lines for both historical and dynamic data result in upward forecasts.

Our global index (DWI) functions as a "market index", while the US index serves as a "risky asset" within the socioeconomic well-being index market. Thus, we aim to determine the "beta" in the capital asset pricing model (CAPM) (Fama and French 2004) for the US index. If more countries are involved, we seek an analog of the Fama–French three-factor model (Fama and French 1992, 1993) for the socioeconomic well-being index market. Essentially, we want to identify which countries have the most significant impact on global well-being.

Jensen's alpha (Jensen 1968) shows the average return on a portfolio or investment above or below the portfolio benchmark.[6] We apply Jensen's alpha to determine the maximum possible return on each country's DWI and the performance of each country's DWI compared to the global DWI.

Table 5 presents Jensen's alpha values for each country's DWI. The historical indices have nonsignificant alphas that are close to zero, indicating that there is no advantage compared to the global DWI. In contrast, the dynamic indices yield different results. South Africa and India show positive alphas (0.5359 and 0.5306, respectively), suggesting that their DWIs offer higher risk-adjusted returns. Japan, Germany, and the US exhibit statistically significant negative alphas, indicating underperformance compared to the global DWI. The UK, Australia, and China's returns are roughly similar to the global DWI.

**Table 5.** Jensen's alpha CAPM for historical and dynamic indices.

| Country | Historical Index | Dynamic Index |
|---|---|---|
| Japan | −0.048 | −0.5247 |
| Germany | 0.0087 | −0.2615 |
| US | −0.0075 | −0.2264 |
| UK | 0.0232 | 0.1487 |
| Australia | 0.0242 | 0.189 |
| China | 0.0558 | 0.1951 |
| Brazil | 0.0198 | 0.4666 |
| India | 0.0894 | 0.5306 |
| South Africa | 0.0404 | 0.5359 |

## 5. Efficient Frontier of the Markets of Countries' Well-Being Risk Measures

In accordance with Markowitz's (1952) method, the goal of portfolio optimization is to identify the daily set of weights $w$ that reduces the return risk of the portfolio (for that day) subject to a desired expected return ($r_p$). The targeted return value indicates the investor's risk tolerance: the higher the value, the more risk the investor is willing to accept. Mean-variance and mean-CVaR optimization (Uryasev and Rockafellar 2001) have the goal of minimizing the portfolio variance $\sigma_p$ and the portfolio CVaR, denoted by $CVaR_{p,\alpha}$, subject to a preferred expected return by using the variance and CVaR as the risk measure.

Consider a portfolio consisting of n risky assets with daily return values $r(t) = (r_1(t), r_2(t), \dots, r_n(t))$, with a portfolio mean and standard deviation $\bar{r} = (\bar{r}_1, \bar{r}_2, \dots, \bar{r}_n)$ and $(\sigma_1, \sigma_2, \dots, \sigma_n)$, respectively:

$$minimize\ w'\sigma_p w \text{ subject to } \bar{r}w = r_p \text{ and } \sum_{i=1}^{n} w_i = 1. \tag{17}$$

Because the target return varies, the best solution for ($\sigma_p, r_p$) results in a hyperbola curve known as the "efficient frontier" (EF), which is the region of the portfolio frontier where the projected mean returns exceed $r_p$. Consider a portfolio consisting of $n$ risky assets with daily return values $r_p$, with the mean of the expected risk-adjusted returns denoted by $E(r_p)$ and the risk measure denoted by $V(r_p)$. The portfolio optimization can be summarized as follows:

$$\min_{w}\left(-\gamma E(r_p) - (1-\gamma)V(r_p)\right) \text{ subject to } \sum_{i=1}^{n} w_i = 1, \tag{18}$$

where the risk-aversion parameter $\gamma \in [0, 1]$ determines the positions along the EF, with $\gamma = 0$ corresponding to the minimum-risk portfolio.

For each country's DWI index, the optimization applied to the ensemble of the target returns produces an EF by considering the variance, $CVaR_{p,0.05}$, and $CVaR_{p,0.01}$ risk measures to contrast the effects of the central risk and tail risk on optimization and since the standard deviation is not a coherent risk measure. The DWI dynamic and historical indices referred to in Section 2.1 are used to illustrate the EFs of each country.

Figures 4 and 5a,b plot the EFs computed for the US dynamic and historical DWIs. The mean variance risk measure was used to generate efficient borders for the US dynamic and historical DWIs. We used the set of equally spaced values $\gamma = 0, 0.01, \dots, 0.99$ to plot each EF. Historical EFs are short since we have only 30 historical data points. We expect a long EF for simulated data. The standard deviation increased in the dynamic EF from 0.05 to 0.80 and in the same frontier for historical data from 0.08 to 0.34. The growth in $E(rp)$ for the dynamic and historical DWIs is consistent with the same observation; however, the increases are less evident for the EF.

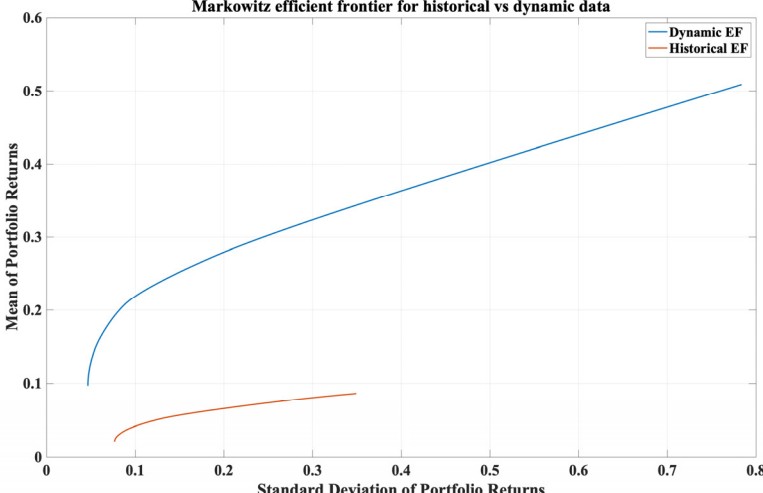

**Figure 4.** Markowitz efficient frontier.

Figure 6a,b reproduce Figure 4 for the tail risk measures $CVaR_{p,0.05}$ and $CVaR_{p,0.01}$. There are overall qualitative similarities in the behaviors of the CVaR EFs compared to the mean-variance EFs. While the EFs change "smoothly" and are convex, the variation in the behavior of the EFs is more pronounced under the CVaR risk measures compared to the mean-variance EF.

In Figures 7 and 8, we show the optimal portfolio weights on the EFs of the mean-variance and CVaR optimizations. For CVaR historical portfolio EFs, India and China's weights increase with increasing standard deviation, emphasizing that these countries are at high risk, along with the US. Conversely, the dynamic portfolio EFs highlight that Brazil, China, and India show higher standard deviation levels. Additionally, the UK and South Africa become focal points at high CVaR risk levels, suggesting their safety in the event of a significant increase in the DWIs.

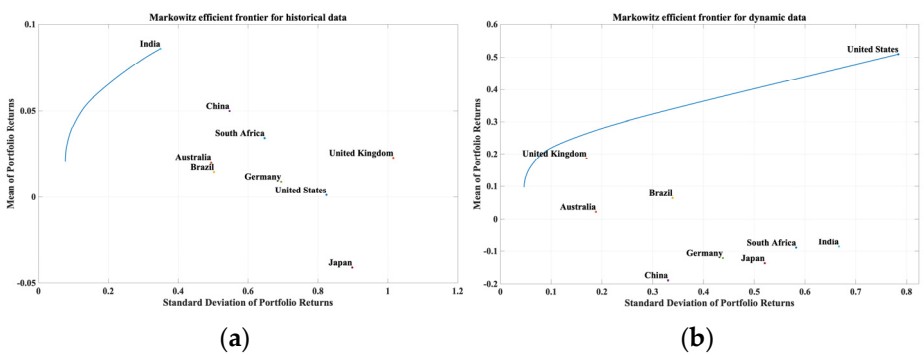

**Figure 5.** Markowitz efficient frontier: (**a**) historical indices and (**b**) dynamic indices.

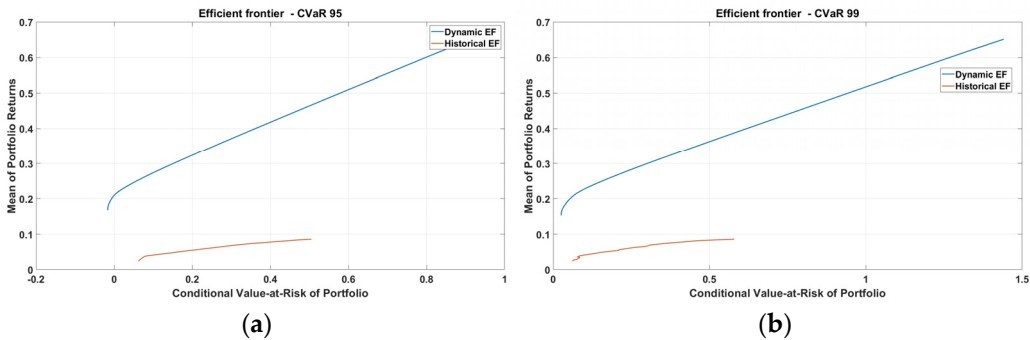

**Figure 6.** Conditional value-at-risk portfolio optimization: (**a**) $CVaR_{p,0.05}$ and (**b**) $CVaR_{p,0.01}$ EFs.

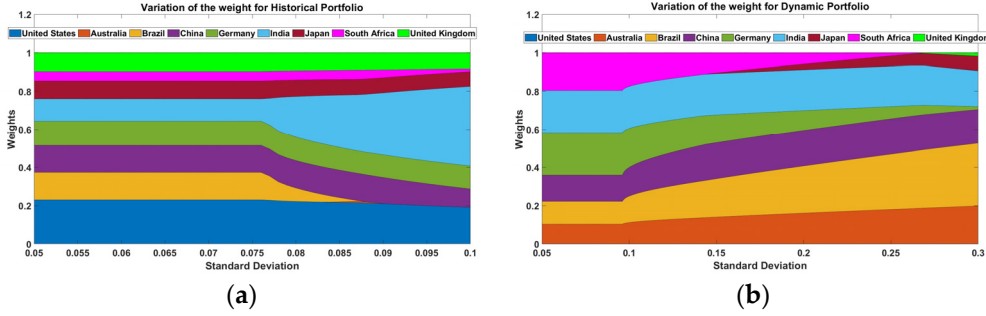

**Figure 7.** Variation of the weight composition of the Markowitz optimal portfolios along each efficient frontier (as a function of standard deviation): (**a**) historical portfolio and (**b**) dynamic portfolio.

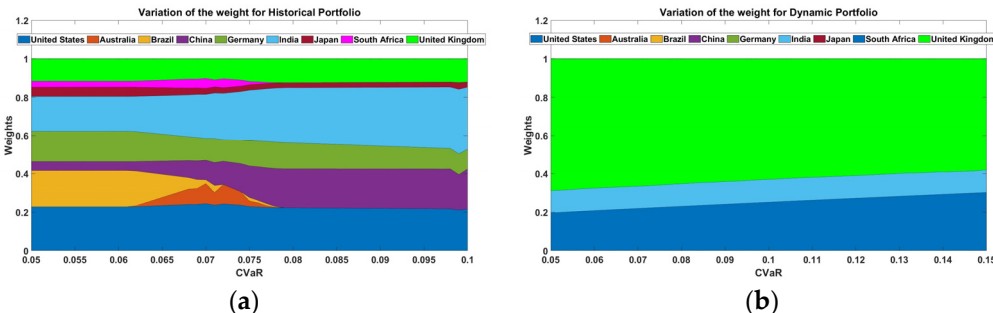

**Figure 8.** Variation of the weight composition of the CVaR$_{p,\alpha}$ optimal portfolios along each efficient frontier (as a function of $\alpha$): (**a**) historical portfolio and (**b**) dynamic portfolio.

*Efficient Frontier and Risk Measures of the Market for Countries with High GDPs*

Economic rankings have evolved over time, and the nominal GDP/capita is a key metric for assessing national wealth. It is intriguing to examine the portfolio of socioeconomic well-being indices for the countries with the largest GDPs and compare them to the portfolios of all countries. As of 2022, the top four nations by nominal GDP were the US (USD 20.89 trillion), China (USD 14.72 trillion), Japan (USD 5.06 trillion), and Germany (USD 3.85 trillion), according to recent World Bank data. In this section, we analyze the portfolios of high-GDP countries; this is akin to the previous section focusing on the top four GDP nations. We employ optimization to create EFs, considering the variance, CVaRp at 0.05, and CVaRp at 0.01 risk measures to assess the effects of central and tail risk on optimization. We also introduce dynamic indices, as explained in Section 2.1, for these four countries, utilizing these new indices to illustrate their respective EFs.

Figure 9a,b display the EFs for the largest-GDP countries, comparing them to the EFs for all countries using mean-variance risk measures for both historical and dynamic DWI portfolios. The historical EF is relatively short due to limited historical data (30 data points), while the simulated data are expected to produce a longer EF. In the high-GDP countries' historical EF, the standard deviation increased from 0.16 to 0.55, and for all countries, it increased from 0.07 to 0.34. Interestingly, the growth in E($r_p$) for the DWI of all countries is higher than it is for high-GDP countries at the same risk level. This suggests that investing in high-GDP countries' DWI carries a higher risk for the same expected return compared to investing in the DWI from all countries. Qualitatively, the dynamic EFs for all countries and high-GDP countries exhibit similar behaviors, but the historical EFs for high-GDP countries display smoother, convex changes, while all countries' EFs show more pronounced fluctuations.

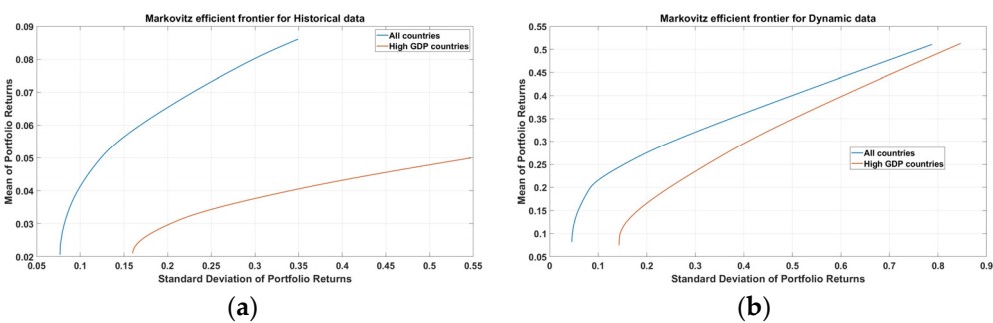

**Figure 9.** Markowitz efficient frontier for all countries and countries with high: (**a**) historical indices and (**b**) dynamic indices.

Figure 10a,b reproduce Figure 9 for the tail risk measures CVaR$_{p,0.05}$ and CVaR$_{p,0.01}$. The behaviors of the CVaR's EFs of the DWI for countries with high GDPs and those for all countries are comparable. Again, it can be seen that the risk of investing in the DWI for all countries is less than the risk for countries with high GDPs at the same expected

return. The variation in the behavior of the EFs is more pronounced under the $\text{CVaR}_{p,0.01}$ risk measure.

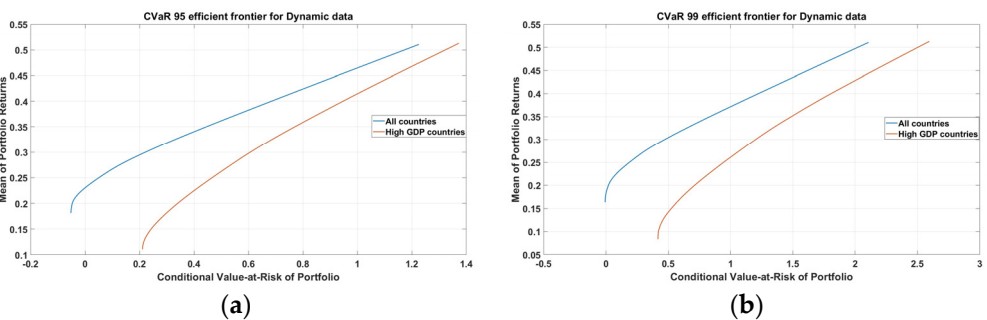

**Figure 10.** Comparing conditional value-at-risk portfolio optimization for dynamic indices: (**a**) $\text{CVaR}_{p,0.05}$ and (**b**) $\text{CVaR}_{p,0.01}$ efficient frontiers.

## 6. Pricing Options on Socioeconomic Well-Being Indices

In this section, we construct a financial model for pricing DWI options. Traditional methods like the Black–Scholes–Merton model, binomial option pricing, trinomial trees, Monte Carlo simulations, and finite difference models have been used to calculate option prices (Hull 2017; Duffie 2010). However, due to the presence of heteroskedasticity and heavy-tailed distributions in DWI returns, we avoid using the Black–Scholes–Merton model for DWI options. Instead, we rely on a discrete stochastic volatility-based model, specifically Duan's (1995) discrete-time GARCH approach with NIG innovations. This dynamic model provides an accurate pricing performance in a volatility-based framework. We provide DWI option prices, catering to institutional investors seeking to incorporate an additional socioeconomic dimension into their risk-adjusted portfolios and addressing potential mispricing events.[7]

In the standard GARCH(1,1) model, Blaesild (1981) defines $R_t$ for a given $F_{t-1}$, as distributed on a real-world probability space (P), as follows:

$$R\_t \sim NIG\left(\lambda, \frac{\alpha}{\sqrt{a_t}}, \frac{\beta}{\sqrt{a_t}}, \delta\sqrt{a_t}, r_t' + m_t + \mu\sqrt{a_t}\right), m_t = \lambda_0\sqrt{a_t} - \frac{1}{2}a_t. \tag{19}$$

Gerber and Shiu (1994) describe the traditional approach to determining an equivalent martingale measure to obtain a constant option price. According to these authors, the Esscher transformation is the conventional method for deriving an equivalent martingale measure to obtain a constant option price. Since the moment-generating function of the NIG distribution has an exponential form, the probability density of $R_t$ is transformed into the risk-neutral probability density using the Esscher transform.

Chorro et al. (2012) discovered that $R_t$ for a given $F_{t-1}$ is distributed on the risk-neutral probability ($Q$) using the Esscher transformation as follows:

$$R\_t \sim NIG\left(\lambda, \frac{\alpha}{\sqrt{a_t}}, \frac{\beta}{\sqrt{a_t}} + \theta_t, \delta\sqrt{a_t}, r_t' + m_t + \mu\sqrt{a_t}\right), \tag{20}$$

where $\theta_t$ is the solution to $MGF(1 + \theta_t) = MGF(\theta_t)\,e^{rt'}$, and $MGF$ is the conditional moment-generating function of $R_{t+1}$ given $F_t$.

Using Monte Carlo simulations, we construct future values of the DWI to price its call and put options as follows:

1. Fit GARCH(1,1) with NIG innovations to $R_t$ and forecast $a_1^2$ by setting $t = 1$;
2. Repeat steps (a)–(d) for $t = 3, 4, \ldots, T$, where $T$ is the time to maturity of the DWI call option from $t = 2$:

   (a) Estimate the model parameter $\theta_t$ using $MGF(1 + \theta_t) = MGF(\theta_t)\,e^{rt'}$, where $MGF$ is the conditional moment-generating function of $R_{t+1}$ given $F_t$ on P;

(b) Find an equivalent distribution function for $\epsilon_t$ on $Q$;

(c) Generate the value of $\epsilon_{t+1}$ under the assumption $\epsilon_t \sim NIG(\lambda, \alpha, \beta, \sqrt{a_t}\,\theta_t + \delta, \mu)$ on $Q$;

(d) Compute the values of $R_{t+1}$ and $a_{t+1}$ using a GARCH(1,1) model with NIG innovations.

3. Generate future values of $R_t$ for $t = 1, \dots, T$ on $Q$, where $T$ is the time to maturity. Recursively, future values of the DWI are obtained as follows:

$$DWI_t = \exp(R_t) \cdot DWI_{t-1}. \tag{21}$$

4. Repeat steps 2 and 3 10,000 ($N$) times to simulate $N$ paths to compute future values of the DWI.

Then, for a specific strike price $K$, the approximate future values of the DWI at time $t$ are the average of the DWIs, and this price is used to determine the price of the call option ($\hat{C}$ and $\hat{P}$, respectively):

$$\hat{C}(t, T, K) = \frac{1}{N} e^{-r'_t(T-t)} \sum_{i=1}^{N} max(DWI_T^{(i)} - K, 0), \tag{22}$$

$$\hat{P}(t, T, K) = (t, T, K) = \frac{1}{N} e^{-r'_t(T-t)} \sum_{i=1}^{N} max(K - DWI_T^{(i)}, 0) \tag{23}$$

Call option pricing ($\hat{C}$) can help investors in planning to purchase our socioeconomic well-being indices at a predetermined strike price within a predetermined time frame (time to maturity).

Figure 11a shows call and option prices for the DWI, depending on the moneyness ($S/K$) and time to maturity ($T$). As the strike price rises, DWI call option prices slightly decrease, while an increase in the maturity time signifies a rise in DWI prices, reflecting increased population happiness within the financial market context. In Figure 11b, we display selling prices for index shares, explaining the relationship between put option prices ($\hat{P}$), the strike price ($K$), and the time to maturity ($T$). Put option prices are lower than call options with the same moneyness and maturity time, but they increase with higher strike prices, suggesting a potential linear relationship. The implied volatility, a reliable indicator of future volatility, is shown in Figure 12 for the DWI; it was constructed using the market values of call option contracts as proxies for upcoming event expectations. The implied volatility is determined based on the time to maturity ($T$) and moneyness ($M = S/K$). During intense market stress, an inverted volatility "grin" may appear on the observed volatility surface, with the highest implied volatilities corresponding to increasing moneyness.

Implied volatilities are higher for lower-strike-price options than for higher-strike-price options due to the downward-sloping volatility skew. As the maturity approaches 16 years, the implied volatilities tend to stabilize. These option prices are better suited for hedging than speculation, serving a role akin to that of portfolio insurance.

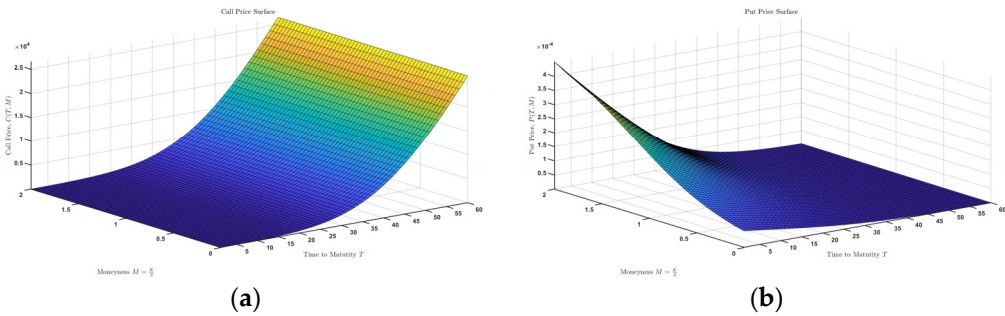

(**a**)                      (**b**)

**Figure 11.** Option prices for the US DWI at time t with varying strike prices $K$ using a GARCH(1,1) model with NIG innovations: (**a**) call prices and (**b**) put prices.

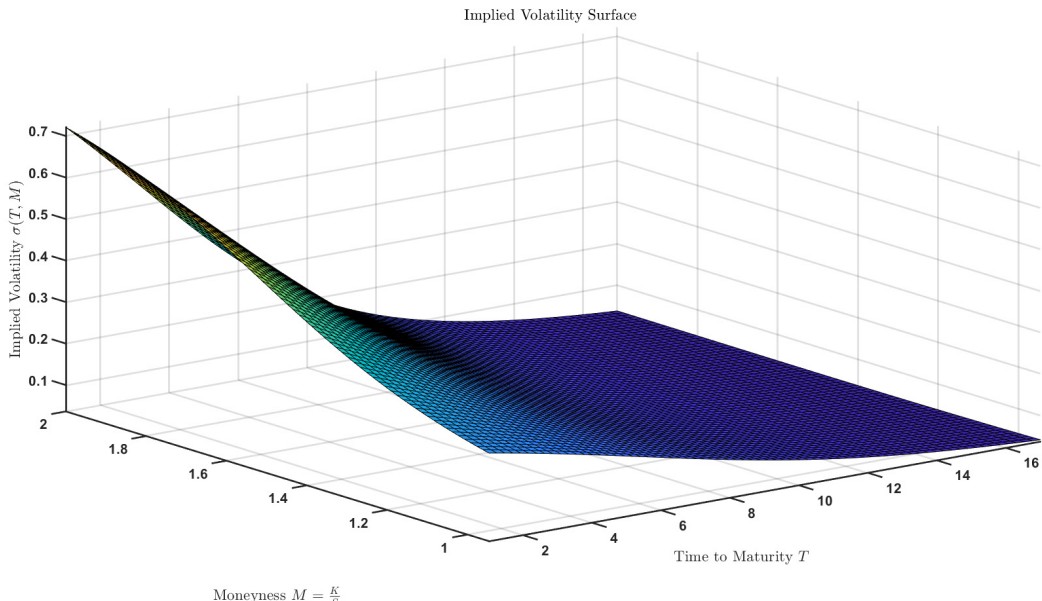

**Figure 12.** Implied volatilities of the US DWI based on the time to maturity ($T$) and moneyness ($M = S/K$) using a GARCH(1,1) model with NIG innovations.

## 7. Discussion and Conclusions

We introduced incorporating the well-being of the entire country into risk-return profiles for institutional investors. We then established a financial market for the socioeconomic well-being indices of nine countries, integrating eight world development indicators. This model enables the financial industry to assess, manage, and hedge against potential adverse index movements globally. We introduced a novel financial market framework for these indices, fostering active financial sector engagement in national well-being. Our approach involves risk assessment using dynamic asset pricing theory and the creation of hedging instruments, empowering institutional investors to trade these indices and establish insurance funds against adverse movements. This provides a new market for securities that protect against declines in social welfare, much like how traditional financial instruments protect against economic downturns.

While ESG financial markets address socioeconomic issues, our focus is broader, encompassing national citizens' well-being. We have offered option prices on these indices as insurance instruments against future index downturns. These findings help in estimating the funds required to enhance global well-being, utilizing financial contracts for insurance against index declines. Developing an early-warning system for global socioeconomic index downturns requires additional data. Our indices are not tradable, but introducing exchange-traded funds (ETFs) would involve replicating their dynamics with fixed-income portfolios through asset liability management. For instance, an ETF replicating the US DWI could use put options as insurance instruments if the US DWI declined in 2024, ensuring that investors receive the put value on the ETF mirroring the US DWI.

Based on the data availability in IBRD (2022), we considered only using data from the US, Australia, Brazil, China, Germany, India, Japan, South Africa, and the UK as representative countries from all the continents, and extracted reported data from between 1990 and 2020. Since this provides a complete data set, missing data imputation is minimal, which minimizes the statistical error due to missing data imputation.

Our proposed method for constructing indices of socioeconomic well-being is not limited to the nine countries discussed in this paper. A similar portfolio analysis will be performed for various regions in the world based on geographical and economic criteria. This allows for addressing the following problems: which individual or group of countries contribute to the world's wellbeing? And to what degree? We will extend these financial management principles to construct a financial market of socioeconomic well-being for

the organization of the petroleum exporting countries (OPEC) by developing the OPEC DWI. Then, we will check the contribution of each OPEC country to the OPEC DWI and determine what would be the effect if a country leaves or joins OPEC. We believe these findings will help the financial industry, which works with world organizations, to address the potential socioeconomic issues related to the well-being of these societies.

**Author Contributions:** Conceptualization, S.R.; methodology, F.J.F. and S.R.; software, T.V.M. and A.S.; formal analysis, T.V.M., A.S. and S.R.; investigation, T.V.M. and A.S.; data curation, T.V.M. and A.S.; writing—original draft preparation, T.V.M. and A.S.; writing—review and editing, T.V.M., A.S., F.J.F. and S.R.; visualization, T.V.M. and A.S.; supervision, S.R.; project administration, S.R. All authors have read and agreed to the published version of the manuscript.

**Funding:** This research received no external funding.

**Data Availability Statement:** The dataset used in this study is publicly available from the World Bank, accessible through the following link: https://datatopics.worldbank.org/world-development-indicators/ (accessed on 7 December 2023). All relevant data supporting our reported results can be accessed in this publicly archived dataset. We confirm our commitment to data transparency and appreciate the opportunity to provide this information.

**Conflicts of Interest:** The authors declare no conflicts of interest.

## Notes

[1] The study, "Report by the Commission on the Measurement of Economic Performance and Social Progress", was led by Joseph Stiglitz, Amartya Sen, and Jean-Paul Fitoussi (see Stiglitz et al. 2009).

[2] The Basel II Accord requires 10,000 scenarios in the generation of future portfolio returns to properly assess the tail risk portfolio of returns (Orgeldinger 2006; Jacobson et al. 2005).

[3] For the Ljung-Box test, please refer to Ljung and Box (1978).

[4] Regarding the CVaR as a coherent risk measure, please refer to Acerbi and Tasche (2002).

[5] The historical regression assumes iid dependent variables. However, our econometric analysis shows that the dependent variables form a time series with characteristics quite different from white noise. The log returns of the socioeconomic well-being indices display heavy-tailed marginal distributions and volatility clustering. Thus, it is essential to employ a time series forecast, as demonstrated above, and conduct OLS and RR regressions on a sample of $S = 10,000$ Monte Carlo iid scenarios.

[6] In real financial markets, Jensen's alpha is generally nonzero, as they often operate in a pre-equilibrium state with price fluctuations (Soros 2015).

[7] For more information about discrete stochastic volatility-based models, refer to, for example, Duan (1995), Barone-Adesi et al. (2008), and Chorro et al. (2012).

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
