# Peer review of "The Financial Market of Indices of Socioeconomic Well-Being"

_jrfm, doi:10.3390/jrfm17010035_

Round 1

Reviewer 1 Report

Comments and Suggestions for Authors

Review of scientific article

“The Financial Market of Indices of Socioeconomic Well-Being”

Authors: Thilini V. Mahanama, Abootaleb Shirvani, Svetlozar Rachev and Frank J. Fabozzi

This scientific article very interesting! The hard work and very good knowledge is evident when you read the content.

I admire the writing style, the subject matter, the advanced knowledge of financial econometrics applied to indicators of socioeconomic well-being, and the idea of considering the financial market as a place to consider them.

Congratulations to the authors!

I have a few small observations:

Row 142:  I think is better to specify GDP/capita instead of GDP.

Row 178: The title of subsection 2.2. “Financial and econometric modeling of socioeconomic well-being indices” is better read “Econometric financial modeling of socioeconomic well-being indices”.

Row 287: the superscript 4, before comma, “being a coherent risk measure,4” – “being a coherent risk measure4, … ” – I am not sure ….

Table 5 should be comprised in Section 4 whose title “Regression and Jensen’s Alphas of the Socioeconomic Well-Being Indices ….” refers exactly to its content (Jensen’s Alphas). Then Table 5 may be addressed in Section 5.

Both in Table 4 and Table 5 I suggest to authors to specify the p-value level of coefficients, using */**/*** - for the probabilities 10%/ 5%/ 1% - specified below each table.

In Figure 5, it is difficult to identify the place of countries using different marker colors. I suggest if it is possible to use the label of each case - if the software allows it.

In section 2.1 is not mentioned if the indicators GNI/capita and GDP/capita are in constant prices. Are comparable the terms of these time series? Which in the unit measure of Foreign Direct Investment (FDI) – as % in GDP or in other unit measure?

Thank you!

I wish you best of luck!

Merry Christmas!

Author Response

Dear Reviewer,

Thank you very much for your suggestions on improving our manuscript. We have addressed them in the manuscript and listed them below:

Suggestion

Action

Row 142: I think it is better to specify GDP/capita instead of GDP.

Replaced GDP/capita whenever necessary

Row 178: The title of subsection 2.2. “Financial and econometric modeling of socioeconomic well-being indices” is better read “Econometric financial modeling of socioeconomic well-being indices”.

Addressed

Row 287: The superscript 4, before the comma, “being a coherent risk measure,4” – “being a coherent risk measure4, … ”

Replaced with measure,4

Table 5: Should be comprised in Section 4 whose title “Regression and Jensen’s Alphas of the Socioeconomic Well-Being Indices ….” refers exactly to its content (Jensen’s Alphas). Then Table 5 may be addressed in Section 5.

Table 5 is available in Section 4

Tables 4 and 5: I suggest specifying the p-value level of coefficients, using // - for the probabilities 10%/ 5%/ 1% - specified below each table.

Figure 5: In Figure 5, it is difficult to identify the place of countries using different marker colors. I suggest if it is possible to use the label of each case - if the software allows it.

Addressed

Section 2.1: In section 2.1, it is not mentioned if the indicators GNI/capita and GDP/capita are in constant prices. Are comparable the terms of these time series?

Which is the unit measure of Foreign Direct Investment (FDI) – as % in GDP or in other unit measure?

GNI/capita and GDP/capita are taken as yearly data over the duration of 2011-2022 as the other indicators used in this study

FDI is in the current USD. We mentioned this in line 142:

foreign direct investment (in current USD)

Reviewer 2 Report

Comments and Suggestions for Authors

The article is devoted to developing a financial market model for well-being indices. The authors use advanced financial econometrics and dynamic asset pricing methodologies as instrumental methods.

After reading and reviewing this paper, I think it has potential for publication, but the authors should revise the comments below to improve the research soundness:

1.       The authors have constructed socioeconomic well-being indices for nine countries, but the rationale for selecting these specific countries for modeling is not sufficiently explained.

2.       The inclusion of eight key world development indicators is crucial for understanding the context of the study. It is recommended that the authors provide a thorough explanation for selecting these specific indicators, outlining their relevance to the research objectives.

3.       The absence of descriptive statistics for the entire dataset is a notable gap. The authors need to include comprehensive descriptive statistics, not only for the US data presented in Figure 1 but also for all countries considered in the study. The paper would benefit from visualizations of statistical data from other countries as well. Including comparative visualizations will help readers grasp variations and trends across different regions.

4.       The authors should explain the use of the data normalization method. Namely, why formula 1 is used, and, for example, not a linear transformation formula; why are the same weights coefficients used, etc.

5.       To strengthen the paper's contribution to the field, the authors should compare their results with findings from other relevant studies.

6.       The "Discussion and Conclusion" section should be expanded to include a more thorough analysis of the results obtained. The authors should discuss the implications of their findings about existing literature and real-world applications. Additionally, a clear indication of the direction for future research will contribute to the paper's overall significance.

Author Response

Dear Reviewer,

Thank you very much for your suggestions on improving our manuscript. We have addressed them in the manuscript and listed them below:

1.       The authors have constructed socioeconomic well-being indices for nine countries, but the rationale for selecting these specific countries for modeling is not sufficiently explained.

We addressed it under the discussion in Section 7 and lines 143-144.

Based on the data availability in IBRD (2022), we selected these nine countries from all the continents and extracted reported data from between 1990 and 2020.

Since this provides a complete data set, missing data imputation is minimal, which minimizes the statistical error due to missing data imputation.

We’ll consider all countries in future work will implement a fill-in methodology for missing data.

2.       The inclusion of eight key world development indicators is crucial for understanding the context of the study. It is recommended that the authors provide a thorough explanation for selecting these specific indicators, outlining their relevance to the research objectives.

We considered the most representative indicators related to well-being from the word development indicators recommended by the word bank. Then, we chose these eight key independent indicators avoiding collinearity and consider their availability of data from 2011-2022.

3.       The absence of descriptive statistics for the entire dataset is a notable gap. The authors need to include comprehensive descriptive statistics, not only for the US data presented in Figure 1 but also for all countries considered in the study. The paper would benefit from visualizations of statistical data from other countries as well. Including comparative visualizations will help readers grasp variations and trends across different regions.

Considering the length of the manuscript, we chose the US as a representative for all the countries used for this study and included figures on descriptive statistics only for the US. We can include comprehensive descriptive statistics for other countries in an appendix as provided in our preprint available in ResearchGate.

4.       The authors should explain the use of the data normalization method. Namely, why formula 1 is used, and, for example, not a linear transformation formula; why are the same weights coefficients used, etc.

We addressed them as follows in line 156:

In order to compare the significance of factors with respect to countries, we normalize each country’s indicator, F(k,l), using the corresponding indicator of all the countries (F(k,l), l = 1,··· ,L = 9) as follows:                    

5.       To strengthen the paper's contribution to the field, the authors should compare their results with findings from other relevant studies.

Our research uniquely introduces a financial market for the socioeconomic well-being indices consistent with dynamic asset pricing theory.

6.       The "Discussion and Conclusion" section should be expanded to include a more thorough analysis of the results obtained. The authors should discuss the implications of their findings about existing literature and real-world applications. Additionally, a clear indication of the direction for future research will contribute to the paper's overall significance.

Please find the modifications under Section 7.

Reviewer 3 Report

Comments and Suggestions for Authors

The article presents a high level of research and them visualization. It was a great pleasure to have the opportunity to become acquainted with your research. 

The manuscript employs appropriate econometric methods, such as ARCH and GARCH, for the research problem. The use of various methods confirms the robustness of the results.  The models appear to be estimated correctly. 

The results bring interesting conclusions that can be applied to passive ETFs and other relevant studies. The results bring interesting conclusions that can be applied to passive ETFs and other relevant studies. Therefore, they have practical and scientific applications.

To improve the manuscript, it is recommended to expand the discussion to include references to similar studies. Rebuilding the abstract to emphasize the purpose of the study and the results may improve the dissemination of the obtained results by search engines.

Author Response

Dear Reviewer,

Thank you very much for your suggestions on improving our manuscript. We have addressed them in the manuscript and listed them below:

It is recommended to expand the discussion to include references to similar studies.

Please find the modifications under Section 7.

Rebuilding the abstract to emphasize the purpose of the study and the results may improve the dissemination of the obtained results by search engines.

Please find the modified abstract.